# Socioeconomic Disparities in Childhood Vaccination Coverage in the United States: Evidence from a Post-COVID-19 Birth Cohort

**DOI:** 10.3390/vaccines13121256

**Published:** 2025-12-18

**Authors:** Xiaoyang Lv, Antong Long, Yansheng Chen, Hai Fang

**Affiliations:** 1School of Public Health, Peking University, Beijing 100191, China; lvxiaoyang85@bjmu.edu.cn (X.L.); antong@stu.pku.edu.cn (A.L.); cys.alan@stu.pku.edu.cn (Y.C.); 2China Center for Health Development Studies, Peking University, Beijing 100191, China; 3Key Laboratory of Health System Reform and Governance (Peking University), National Health Commission, Beijing 100191, China

**Keywords:** childhood immunization, vaccination disparities, socioeconomic factors, COVID-19

## Abstract

**Background**: Childhood immunization is one of the most effective public health strategies for reducing morbidity and mortality from vaccine-preventable diseases. Although overall vaccination coverage in the United States remains high, disparities persist across socioeconomic and healthcare access groups. Understanding these disparities is particularly important in the post-COVID-19 era, when increased vaccine hesitancy may threaten progress in maintaining equitable coverage. **Materials and Methods**: We analyzed data from the National Immunization Survey–Child (NIS-Child), focusing on U.S. children aged 19–35 months in 2023, corresponding to cohorts reaching this age during or after the COVID-19 pandemic. The primary outcome was receipt of the up-to-date combined 7-vaccine series (4:3:1:3:3:1:3: ≥4 doses of DTaP, ≥3 doses of polio, ≥1 dose of measles-containing vaccine, full Hib series, ≥3 doses of hepatitis B, ≥1 dose of varicella, and ≥3 doses of PCV). Logistic regression models were used to estimate associations between vaccination coverage and key explanatory variables: household income-to-poverty ratio, maternal education, health insurance type, and provider facility type, controlling for demographic and regional covariates. Disparities were quantified using concentration indices (CIs). **Results**: Among children in the analytic sample, overall coverage for the 7-vaccine series was only 78.5%. Nonetheless, disparities were evident. Children from households with lower income-to-poverty ratios (<1 × FPL: OR = 0.44, 95% CI = 0.37–0.53; 100–200%: OR = 0.66, 95% CI = 0.56–0.79), those covered by Medicaid (OR = 0.54, 95% CI = 0.45–0.64), other insurance (OR = 0.48, 95% CI = 0.37–0.61), or uninsured (OR = 0.27, 95% CI = 0.18–0.42), and those whose mothers had lower educational attainment (<12 years: OR = 0.35, 95% CI = 0.28–0.44) had significantly lower odds of being up-to-date. Similar associations were observed across specific vaccines. Unadjusted CIs for income-to-poverty ratio (0.04, *p* < 0.01), maternal education (0.04, *p* < 0.01), health insurance (0.03, *p* < 0.01), and provider type (0.03, *p* < 0.01) decreased but remained statistically significant after adjustment (0.02, 0.02, 0.01, and 0.02, respectively; all *p* < 0.01). No significant disparities were found by census region or race/ethnicity. **Discussion**: Despite relatively high overall vaccination coverage among U.S. children born during and after the COVID-19 pandemic, disparities by socioeconomic and healthcare access factors persisted. However, the absolute magnitude of these disparities was very small (concentration indices ≤ 0.04). These findings suggest that while inequities remain statistically measurable, their scale is limited in absolute terms. Targeted efforts to address income, insurance, maternal education, and provider-related barriers will be important to sustain equitable immunization coverage in the post-pandemic era.

## 1. Introduction

Childhood immunization remains one of the most effective public health interventions to reduce morbidity and mortality from vaccine-preventable diseases across the world [1]. In the United States (US), the Centers for Disease Control and Prevention (CDC) recommends a series of vaccines by age 19–35 months, including DTaP, polio, measles-containing vaccine (MCV), Hib, hepatitis B, varicella, and pneumococcal conjugate vaccine (PCV), many of which achieve high national coverage [2]. The childhood vaccination in the US leads to significant health and economic returns. Among approximately 117 million children born during 1994–2023 in the United States, routine childhood vaccinations are estimated to avert approximately 508 million lifetime cases of illness, 32 million hospitalizations, and 1,129,000 deaths, equivalizing to a net savings of $540 billion in direct costs and $2.7 trillion in societal costs [3]. Nevertheless, maintaining high overall coverage does not preclude the existence of disparities across socioeconomic, insurance, provider, and/or demographic groups [4].

Prior to the COVID-19 pandemic, numerous studies documented associations between lower immunization coverage and socioeconomic factors such as poverty status, type of health insurance, maternal education, and provider facilities in the US [5,6,7,8]. Children insured through Medicaid, uninsured children, or those from low-income households have historically shown lower vaccination rates. Maternal education has also been consistently identified as a strong predictor of childhood vaccine uptake. Provider facility differences—public vs. private, hospital-based vs. private clinics—also influence access, timeliness, and completeness of vaccination.

The COVID-19 pandemic disrupted many aspects of healthcare delivery, including routine pediatric vaccination. A recent USCDC report found that children born in 2020–2021 had lower coverage for nearly all routinely recommended vaccines compared with those born before the pandemic, highlighting a drop in coverage for DTaP and MMR objectives and concerns about recovery of vaccine rates to pre-pandemic levels [9]. In addition, evidence suggests that parental vaccine hesitancy has grown in the US, particularly related to COVID-19 vaccines, and that such hesitancy may extend to routine childhood immunizations [10]. Parents worried about vaccine safety tend to delay or skip non-COVID vaccines as well.

Given these developments, children born during or after the COVID-19 period in the US represent a particularly relevant cohort for assessing whether traditional social determinants of vaccination continue to produce disparities—or whether the pattern of disparities has shifted, reduced, or perhaps widened under pandemic conditions. While global studies have documented long-term gains in routine childhood vaccination coverage since 1980, recent decades have seen stagnating progress and persistent inequities, exacerbated by the COVID-19 pandemic [11]. The pre-pandemic burden of communicable diseases among children and adolescents remained substantial, with disparities closely tied to socioeconomic development [12]. In the US, improving childhood vaccination coverage has been identified as a key lever for enhancing future population health and life expectancy [13]. Monitoring inequities is essential to inform public health strategies aimed at closing gaps and ensuring equitable protection for all children. Building on this background, the present study aimed to examine disparities in routine childhood immunization among US children born during or after the COVID-19 pandemic.

## 2. Methods

### 2.1. Data Source and Study Design

We analyzed data from the 2023 National Immunization Survey–Child (NIS-Child), an annual, nationally representative survey of U.S. children aged 19–35 months [14]. NIS-Child identifies eligible households by random-digit dialing and conducts parent/guardian interviews by telephone; with consent, vaccination histories are verified by providers via mailed questionnaires. All estimates use the provider-verified sampling weights and account for the survey’s stratified, clustered design.

### 2.2. Study Population

We restricted the analytic sample to children with non-missing vaccination outcomes and positive provider-verified weights (PROVWT_C). Survey design variables (STRATUM for strata and SEQNUMHH for primary sampling units) and PROVWT_C were used to define the complex design [15].

### 2.3. Outcomes for Vaccination Coverage

The primary outcome was the standard combined-series measure of up-to-date (UTD), age-appropriate, provider-verified immunization coverage (4:3:1:3:3:1:3), defined as receipts of 7 vaccine types: Diphtheria and tetanus toxoids and acellular pertussis vaccine (DTaP) ≥ 4 doses; polio (IPV) ≥ 3 doses; measles-containing vaccine (MCV) ≥ 1 dose; Hib full series (product-specific); hepatitis B ≥ 3 doses; varicella ≥ 1 dose; and PCV ≥ 3 doses [16]. This measure is coded 1 if up-to-date, and 0 otherwise. As secondary outcomes, we analyzed seven antigen-specific binary indicators corresponding to each component listed above (coded 1 if up-to-date, 0 otherwise).

### 2.4. Key Explanatory Variables for Potential Disparity

We examined six dimensions of explanatory variables for potential disparity:Household income relative to the federal poverty level (FPL), categorized as <1 × FPL, 1–<2 × FPL, and 2–≤3 × FPL (reference).Child’s health insurance: Private (reference), Medicaid, Other Insurance, Uninsured.Education of mother: less than 12 years, 12 years, >12 years but non-college degree, and college degree (reference), four ascending categories (lowest to highest).Usual vaccination provider type: All private (reference), Mixed, All Public, All Hospital, and Military/Other.U.S. region (Census regions): Northeast (reference), Midwest, South, and West.Race/ethnicity: Non-Hispanic White (reference), Non-Hispanic Black, Hispanic, and Other.

Reference groups in these key explanatory variables follow NIS coding conventions and our analysis plan.

### 2.5. Other Covariates

Models adjusted for factors associated with vaccine uptake in NIS-Child and used in our standardization: participation in WIC (ever), child age in months grouped (19–23 as reference, 24–29, and 30–35), child sex (male as reference), maternal age ≥ 30 years (maternal age < 30 years as reference), and maternal marital status (unmarried, or married as reference). Covariate definitions followed the 2023 PUF codebook.

### 2.6. Statistical Analysis

#### 2.6.1. Survey-Weighted Logistic Regression

For each outcome for vaccination coverage, we estimated survey-weighted logistic regression models (linearized variance with strata and PSUs) to obtain odds ratios (ORs) and 95% confidence intervals (CIs) for each key factor, using the reference categories listed above. Because NIS-Child employs complex sampling and weighting, all models used PROVWT_C and accounted for STRATUM and SEQNUMHH. Two-sided tests with α = 0.05 defined statistical significance.

#### 2.6.2. Equity Analysis: Adjusted Concentration Indices

To quantify socioeconomic-related inequality in each vaccination outcome, we computed **Concentration Indices (CI)** for binary variables using the **bounded correction with known limits [0, 1]** as implemented in the Stata **conindex** command [17]. This correction is appropriate for outcomes constrained between 0 and 1 and is widely used in health equity analyses.

Because several disparity dimensions are categorical and nominal (e.g., insurance type, provider type, region, race/ethnicity), we constructed rank variables as follows before estimating each CI. Poverty and maternal education were treated as naturally ordered categories (from low to high). Concentration indices are designed to quantify inequality along a socioeconomic gradient, which requires an ordered rank that reflects relative socioeconomic position. For insurance type, provider type, census region, and race/ethnicity—dimensions that do not have an inherent socioeconomic order—we first established a meaningful ordering to allow concentration indices to be calculated. The income-to-poverty ratio provides a direct, widely used, and interpretable measure of household material resources and socioeconomic advantage, and is therefore well suited as a unifying ordering metric. Individuals were then assigned a relative rank according to this ordering, reflecting their position along the socioeconomic spectrum rather than simple category membership.

For each dimension, we estimated an adjusted CI via indirect standardization. We fit a survey-weighted logistic regression of the outcome on all covariates and all disparity dimensions except the one being studied; obtain the predicted probability and compute the standardized outcome, truncated to [0, 1]; calculate the CI using the dimension-specific rank variable and survey design features. This yields an “inequality net of other factors,” preserving differences attributable to the dimension under study. Unadjusted Cis were also provided for comparison purposes. Sensitivity analyses were conducted by comparing unadjusted and adjusted (standardized) concentration indices across all disparity dimensions examined. All analyses were conducted in Stata (version 17) using survey commands and the conindex package. Documentation for NIS-Child 2023, including variable names and weighting procedures, guided coding and design specification.

### 2.7. Ethical Considerations

NIS-Child public-use data are de-identified and publicly available; analyses of these data are considered exempt from human subjects review.

## 3. Results

A total of 18,032 children with available immunization records in 2023 were included in the study. All percentages reported are weighted to account for the survey’s complex sampling design. Table 1 presents descriptive statistics for the overall sample and stratified by whether the 7-vaccine series was up-to-date. In the overall sample, 14,648 children (78.5%) were up-to-date for the 7-vaccine series. Among individual vaccines, DTaP (≥4 doses) had the lowest coverage (82.2%), while coverage for all other vaccines exceeded 90%. With respect to socioeconomic characteristics, 25.9% of children lived in households with income <1 × FPL, 21.4% between 1 and 2 FPL, and 52.0% above 2 FPL. Regarding health insurance, 43.0% had private insurance only, 47.8% were covered by Medicaid, 6.1% had other insurance only, and 3.0% were uninsured. For maternal education, 11.0% of mothers had fewer than 12 years of education, 26.8% had 12 years, 21.5% had more than 12 years but no college degree, and 40.7% had a college degree. In terms of provider facilities, 54.4% were all private, 12.5% were mixed, 2.7% were military/other, 12.7% were all public, and 17.7% were all hospital-based. By US census region, the largest share of children resided in the South (40.4%). In terms of race/ethnicity, 43.3% were non-Hispanic White, 13.8% non-Hispanic Black, 28.5% Hispanic, and 14.4% other non-Hispanic races or multiple races.

Comparisons between the up-to-date (N = 14,648) and not-up-to-date (N = 2284) groups revealed notable differences. In the not-up-to-date group, coverage for DTaP4 was only 17.2%, and coverage for the other six vaccines ranged between 50 and 60%. Children in this group were more likely to come from lower-income households, less likely to have private insurance only, more likely to have mothers with fewer years of education, and more often received vaccinations at all public or hospital facilities. Differences in distribution by US census region and race/ethnicity were relatively small between the two groups.

Table 2 reports odds ratios (ORs) from logistic regression models assessing associations between key explanatory variables and receipt of the up-to-date 7-vaccine series. In columns (1) through (6), each explanatory variable was entered separately without adjustment for other key explanatory variables or covariates. In these models, children from households with lower income-to-poverty ratios (<1 × FPL, OR = 0.44, 95% CI: 0.37–0.53; 1–<2 × FPL, OR = 0.66, 95% CI: 0.56–0.79), those covered by non-private health insurance (Medicaid, OR = 0.54, 95% CI: 0.45–0.64; other insurance, OR = 0.48, 95% CI: 0.37–0.61; uninsured, OR = 0.27, 95% CI: 0.18–0.42), those whose mothers had fewer years of education (<12 years, OR = 0.35, 95% CI: 0.28–0.44; 12 years, OR = 0.50, 95% CI: 0.42–0.61; >12 years without a college degree, OR = 0.58, 95% CI: 0.48–0.71), and those receiving care from all public providers (OR = 0.49, 95% CI: 0.40–0.61) or all hospital-based providers (OR = 0.58, 95% CI: 0.46–0.72) had significantly lower odds of being up-to-date, compared to those specific references. Regional disparities were observed only for children living in the West (OR = 0.76, 95% CI: 0.63–0.92), compared with those in the Northeast. No significant disparities were detected by race/ethnicity. Column (7) presents results from the fully adjusted model, which included all six key explanatory variables and other covariates. The estimated ORs remained in the same direction and of similar magnitude as in the unadjusted models, suggesting that the associations were robust to adjustment. Although some effect sizes were slightly attenuated, the overall pattern of disparities persisted

Table 3 reports odds ratios (ORs) from logistic regression models examining associations between key explanatory variables and coverage for each of the seven vaccines, after adjusting for covariates. The associations varied somewhat by vaccine but showed consistent patterns. Children from households with lower income-to-poverty ratios (<1 × FPL and 1–<2 × FPL) had significantly lower odds of being up-to-date on DTP4, Pol3, MCV1, Hib3, HepB3, VCV1, and PCV3 compared with those from households with income 2–≤3 × FPL. Similarly, those covered by non-private health insurance (Medicaid, other insurance, or uninsured) had consistently lower odds of up-to-date vaccination across all seven vaccines relative to those with private insurance only. Maternal education below a college degree was also associated with reduced odds of full coverage for each vaccine, with the strongest disparities observed among children whose mothers had less than 12 years of education. Provider type showed a parallel pattern: children receiving care from all public or all hospital-based facilities were less likely to be up-to-date on all seven vaccines compared with those seen exclusively in private facilities. In contrast, census region and race/ethnicity were not significantly associated with vaccination status for any of the seven specific vaccines.

Table 4 presents concentration indices (CIs) for disparities in receipt of the 7-vaccine series, both before and after adjustment of other key explanatory variables and covariates. The unadjusted CI for income-to-poverty category was 0.04 (*p* < 0.01), which decreased to 0.02 (*p* < 0.01) after adjustment. For health insurance type, the CI decreased from 0.03 (*p* < 0.01) in the unadjusted analysis to 0.01 (*p* < 0.01) in the adjusted model. Similarly, unadjusted CIs for maternal education (0.04, *p* < 0.01) and provider facility type (0.03, *p* < 0.01), respectively, declined to 0.02 (all *p* < 0.01) in the adjusted models. By contrast, CIs for census region and race/ethnicity were close to zero and not statistically significant, with the exception of a small unadjusted CI for race/ethnicity (0.01), which lost significance after adjustment. Overall, these findings suggest that socioeconomic factors—including household income, health insurance, maternal education, and provider type—were the main contributors to disparities in childhood vaccination coverage, though the absolute magnitude of these disparities was extremely small. Regional and racial/ethnic differences were very minimal. Table 5 reports unadjusted concentration indices (CIs) for key explanatory variables—including income-to-poverty category, health insurance type, maternal education, provider facility type, census region, and race/ethnicity—across each of the seven specific vaccines. The estimated disparities were also very small, with none exceeding 0.02. The direction and relative magnitude of the concentration indices were consistent with those observed in the adjusted analyses of the 7-vaccine series, indicating that the observed disparities were not sensitive to model specification or outcome definition.

## 4. Discussion

This study used data from National Immunization Survey Child 2023 in the US, and analyzed vaccination data from 18,032 children born after the COVID-19 pandemic to assess disparities in the up-to-date coverage of 7-vaccine series and its individual components. Our findings indicated that overall vaccination coverage remained relatively high in the US, but significant disparities persisted among socioeconomic and healthcare access factors. Consistent with previous research, our study found that children from lower-income households, without private insurance, with mothers possessing fewer years of education, and/or without private immunization providers had significantly lower vaccination rates. Our study extended these findings by demonstrating that these disparities were not only present in overall vaccination rates but also in specific vaccines within the 7-vaccine series, including DTP4, Pol3, MCV1, Hib3, HepB3, VCV1, and PCV3. This underscored the need for targeted interventions to address these inequities. Interestingly, our analysis revealed minimal disparities based on race/ethnicity, while this contrasts with some studies that have identified racial and regional disparities in vaccination coverage. Because vaccination coverage in the US was high overall during this period with the up-to-date coverage rate for most of 7 specific vaccines greater than 90%, concentration indices for binary outcomes bounded on [0, 1] tended to be numerically small even when adjusted ORs were statistically significant—a known property of CI measures near boundary means [18]. Nevertheless, even small absolute disparities can have public health significance if they persist across large populations or over time, potentially leading to clusters of under-immunization and increased outbreak risk in disadvantaged communities.

Our results were consistent with those in the previous studies about childhood vaccination in the US that up-to-date coverages were associated with many socioeconomic factors. Hill et al. reported that among children born in 2020, vaccination coverage was 4–14 percentage points lower among those eligible for the Vaccines for Children (VFC) program compared to non-eligible children, highlighting the impact of socioeconomic status on vaccination uptake [2]. VFC eligibility in the United States was defined as meeting at least one of the following criteria: (1) American Indian or Alaska Native; (2) insured by Medicaid, Indian Health Service (IHS), or uninsured; or (3) ever received at least one vaccination at an IHS-operated center, Tribal health center, or urban Indian healthcare facility. VFC has been very effective in reducing disparities in vaccination coverage among U.S. children [19]. Although the VFC program initiated in 1993 has played a vital role in increasing and maintaining high levels of childhood vaccination coverage, disparities remain [20].

Low vaccination coverage among children living in and near poverty has been a persistent problem in the United States [21]. Vaccination coverage was lower among Black, Hispanic, and American Indian/Alaska Native children [22], and those insured by Medicaid or other nonprivate insurance [23]. Families who had moved since the child’s birth, especially if the mother had high school or lower education, were less likely to have children UTD on the vaccine series [24]. During the 2024–2025 school year, vaccination coverage among kindergartners in the U.S. decreased for all reported vaccines from the year before [25]. Consistent with this established body of evidence, our findings indicate that socioeconomic disparities in vaccination coverage persisted among children born during or after the COVID-19 pandemic. Pre-pandemic analyses of NIS-Child cohorts similarly documented lower vaccination coverage among children living below the poverty level, with reported gaps ranging from 2.7 percentage points for ≥1 dose of MMR to as high as 19.9 percentage points for ≥2 doses of influenza vaccine, depending on the vaccine examined [9].

The persistence of socioeconomic disparities in vaccination coverage has significant public health implications. Our findings suggest that targeted efforts should prioritize children from households with incomes <2 × FPL, those covered by Medicaid or other non-private insurance (or uninsured), those whose mothers have lower educational attainment, and those relying on public or hospital-based vaccination providers. Despite high overall vaccination rates, these disparities contribute to uneven protection against vaccine-preventable diseases, potentially leading to outbreaks in under-immunized communities. Addressing these disparities requires multifaceted strategies, such as implementing community-based programs to educate parents about the importance of vaccination, particularly in underserved areas, expanding access to healthcare facilities that offer vaccinations, especially in rural and low-income urban areas, and strengthening policies that ensure equitable access to vaccines, such as the VFC program, and considering adjustments to address identified gaps.

This study’s strengths include its large, nationally representative sample and the use of provider-verified vaccination data, which enhance the reliability of the findings. However, several limitations should be considered. First, the study’s cross-sectional nature limits the ability to infer causality between socioeconomic factors and vaccination coverage, as we mainly focused on the children who were born during and after COVID-19 pandemic. While efforts were made to ensure accurate data collection, there may be unmeasured factors influencing vaccination status, such as parental attitudes towards vaccines. Second, the study’s findings may not be fully generalizable to populations outside the United States or to children born before the COVID-19 pandemic. Third, our analysis included children with missing data on vaccination outcomes, key explanatory variables, or sampling weights. While this method aligns with NIS-Child analytical guidelines and our sensitivity analysis suggested robustness, the exclusion of these cases may introduce selection bias if the missingness is not random, potentially affecting the generalizability of our findings.

## 5. Conclusions

In conclusion, while overall childhood vaccination coverage in the United States remains high, significant disparities persist, particularly among children from lower socioeconomic backgrounds. These disparities underscore the need for targeted public health interventions to ensure equitable access to vaccines and to maintain high immunization rates across all demographic groups.

## Figures and Tables

**Table 1 vaccines-13-01256-t001:** Descriptive Statistics of Vaccination Coverage and Socioeconomic Characteristics in the 2023 NIS-Child Sample.

Variable	Overall (N = 18,032)		Updated (N = 14,648)		Not Updated (N = 3384)	
Vaccination						
7-vaccine series (4:3:1:3:3:1:3)	14,648	(78.5%)	14,648	(100%)	0	(0%)
DTP4	15,308	(82.2%)	14,648	(100%)	660	(17.2%)
Pol3	16,732	(91.9%)	14,648	(100%)	2084	(62.1%)
MCV1	16,678	(91.8%)	14,648	(100%)	2030	(62.0%)
Hib3	16,467	(90.5%)	14,648	(100%)	1819	(55.7%)
HepB3	16,591	(91.3%)	14,648	(100%)	1943	(59.6%)
VCV1	16,511	(90.9%)	14,648	(100%)	1863	(57.6%)
PCV3	16,688	(91.7%)	14,648	(100%)	2040	(61.2%)
Income to poverty (FPL) category						
<1 × FPL	2883	(25.9%)	2060	(22.8%)	823	(37.0%)
1–<2 × FPL	3307	(21.4%)	2504	(21.0%)	803	(22.7%)
2–≤3 × FPL	11,842	(52.7%)	10,084	(56.1%)	1758	(40.3%)
Health insurance type						
Private only	10,019	(43.0%)	8652	(46.4%)	1367	(30.6%)
Any Medicaid	6237	(47.8%)	4708	(45.6%)	1529	(56.0%)
Other insurance	1355	(6.1%)	1029	(5.7%)	326	(7.8%)
Uninsured	421	(3.0%)	259	(2.3%)	162	(5.6%)
Education of mother						
<12 years	1035	(11.0%)	679	(9.4%)	356	(17.1%)
12 years	2728	(26.8%)	2031	(25.4%)	697	(31.9%)
>12 years, non-college grad	4096	(21.5%)	3197	(21.1%)	899	(23.0%)
College grad	10,173	(40.7%)	8741	(44.2%)	1432	(28.0%)
Provider facility type						
All private	9545	(54.4%)	8150	(56.7%)	1395	(45.4%)
Mixed	2394	(12.5%)	2032	(12.9%)	362	(11.0%)
All military/other	401	(2.7%)	304	(2.7%)	97	(2.7%)
All public	1996	(12.7%)	1466	(11.3%)	530	(18.2%)
All hospital	3387	(17.7%)	2696	(16.4%)	691	(22.7%)
Census region						
Northeast	3232	(15.8%)	2745	(16.5%)	487	(13.6%)
Midwest	4275	(20.5%)	3511	(20.6%)	764	(20.1%)
South	6457	(40.4%)	5186	(40.8%)	1271	(39.0%)
West	4068	(23.2%)	3206	(22.1%)	862	(27.2%)
Race/ethnicity						
Non-hispanic white only	10,492	(43.3%)	8686	(44.1%)	1806	(40.4%)
Non-hispanic black only	1423	(13.8%)	1104	(13.5%)	319	(14.9%)
Hispanic	3440	(28.5%)	2723	(28.0%)	717	(30.1%)
Non-hispanic other + multiple race	2677	(14.4%)	2135	(14.4%)	542	(14.7%)
WIC, ever received	5914	(46.0%)	4547	(44.1%)	1367	(52.8%)
Age category						
19–23 months	5615	(30.9%)	4309	(28.6%)	1306	(39.1%)
24–29 months	5432	(33.8%)	4470	(34.7%)	962	(30.5%)
30–25 months	6985	(35.3%)	5869	(36.7%)	1116	(30.3%)
Sex						
Male	9253	(51.1%)	7518	(51.1%)	1735	(51.2%)
Female	8779	(48.9%)	7130	(48.9%)	1649	(48.8%)
Age of mother						
<30 years	4477	(31.7%)	3403	(30.3%)	1074	(36.8%)
≥30 years	13,555	(68.3%)	11,245	(69.7%)	2310	(63.2%)
Marital status of mother						
Married	13,399	(61.5%)	11,114	(63.2%)	2285	(55.0%)
Never married/widowed/divorced/separated/deceased/living with partner	4633	(38.5%)	3534	(36.8%)	1099	(45.0%)
Number of people in household						
2	441	(3.9%)	353	(3.9%)	88	(4.0%)
3	4471	(23.2%)	3805	(24.4%)	666	(18.5%)
4	6719	(33.7%)	5623	(35.1%)	1096	(28.7%)
5	3510	(19.2%)	2796	(19.4%)	714	(18.5%)
6	1667	(10.1%)	1247	(9.0%)	420	(14.1%)
7	681	(5.1%)	491	(4.6%)	190	(6.9%)
8	543	(4.7%)	333	(3.5%)	210	(9.3%)

Note: (1) 7-vaccine series (4:3:1:3:3:1:3): 4 DTaP/DTP/DT; 3 Polio (IPV/OPV); 1 MMR; 3* Hib (full series per product); 3* HepB; 1 Varicella; 3 PCV. (2) Values are n (column %). Counts are unweighted; percentages are survey-weighted and account for stratification and clustering. Percentages may not total 100% due to rounding. (3) FPL = Federal Poverty Level. Categorized as < 1 × FPL, 1–<2 × FPL, and 2–≤3 × FPL (reference). (4) Other insurance includes CHIP, HIS, military, or other, alone or in combined with private insurance.

**Table 2 vaccines-13-01256-t002:** Odds Ratios for Associations Between Key Socioeconomic Factors and Up-to-Date 7-Vaccine Series Coverage.

Variable	(1)	(2)	(3)	(4)	(5)	(6)	(7)
**Income to poverty (FPL) category**							
<1 × FPL	0.44 (0.37, 0.53)						0.64 (0.46, 0.89)
1–<2 × FPL	0.66 (0.56, 0.79)						0.86 (0.65, 1.15)
2–≤3 × FPL	Ref.						Ref.
**Health insurance type**							
Private only		Ref.					Ref.
Any Medicaid		0.54 (0.45, 0.64)					0.82 (0.62, 1.10)
Other insurance		0.48 (0.37, 0.61)					0.61 (0.46, 0.81)
Uninsured		0.27 (0.18, 0.42)					0.40 (0.25, 0.64)
**Education of mother**							
<12 years			0.35 (0.28, 0.44)				0.58 (0.42, 0.80)
12 years			0.50 (0.42, 0.61)				0.69 (0.54, 0.89)
>12 years, non-college grad			0.58 (0.48, 0.71)				0.71 (0.56, 0.91)
**Provider facility type**							
All private				Ref.			Ref.
Mixed				0.94 (0.73, 1.21)			1.01 (0.79, 1.30)
All military/other				0.80 (0.53, 1.21)			0.93 (0.62, 1.39)
All public				0.49 (0.40, 0.61)			0.67 (0.53, 0.85)
All hospital				0.58 (0.46, 0.72)			0.64 (0.51, 0.80)
**Census region**							
Northeast					Ref.		Ref.
Midwest					0.84 (0.70, 1.02)		0.95 (0.78, 1.17)
South					0.86 (0.71, 1.05)		0.95 (0.77, 1.17)
West					0.67 (0.52, 0.86)		0.67 (0.52, 0.86)
**Race/ethnicity**							
Non-hispanic white only						Ref.	Ref.
Non-hispanic black only						0.83 (0.65, 1.07)	0.94 (0.73, 1.22)
Hispanic						0.85 (0.70, 1.04)	1.23 (1.00, 1.52)
Non-hispanic other + multiple race						0.89 (0.74, 1.09)	0.90 (0.73, 1.11)
**WIC, ever received**							1.30 (1.04, 1.63)
**Age category**							
19–23 months							Ref.
24–29 months							1.57 (1.30, 1.90)
30–25 months							1.73 (1.42, 2.11)
**Sex**							
Male							Ref.
Female							1.02 (0.87, 1.19)
**Age of mother**							
<30 years							Ref.
≥30 years							1.08 (0.90, 1.30)
**Marital status of mother**							
Married							Ref.
Never married/widowed/divorced/separated/deceased/living with partner							0.95 (0.80, 0.93)
**Number of people in household**							0.86 (0.80, 0.93)
Constant	5.09 (4.57, 5.68)	5.55 (4.88, 6.30)	5.76 (5.21, 6.38)	4.80 (4.32, 5.34)	4.43 (3.80, 5.16)	3.99 (3.65, 4.36)	10.25 (6.83, 15.38)

Note: (1) 7-vaccine series (4:3:1:3:3:1:3): 4 DTaP/DTP/DT; 3 Polio (IPV/OPV); 1 MMR; 3* Hib (full series per product); 3* HepB; 1 Varicella; 3 PCV. (2) Odds ratios (ORs) and 95% confidence intervals from survey-weighted logistic regression accounting for stratification and clustering.

**Table 3 vaccines-13-01256-t003:** Odds Ratios for Associations Between Key Socioeconomic Factors and Coverage of Individual Vaccines.

Variable	OR (95% CI)						
	DTP4	Pol3	MCV1	Hib3	HepB3	VCV1	PCV3
Income to poverty (FPL) category							
<1 × FPL	0.66 (0.46, 0.95)	0.53 (0.33, 0.83)	0.99 (0.53, 1.83)	0.53 (0.34, 0.82)	0.74 (0.40, 1.37)	0.99 (0.57, 1.72)	0.55 (0.35, 0.86)
1–<2 × FPL	0.95 (0.70, 1.30)	0.82 (0.545, 1.24)	1.04 (0.63, 1.72)	0.74 (0.51, 1.07)	0.97 (0.59, 1.62)	1.12 (0.71, 1.77)	0.74 (0.49, 1.10)
2–≤3 × FPL	Ref.	Ref.	Ref.	Ref.	Ref.	Ref.	Ref.
Health insurance type							
Private only	Ref.	Ref.	Ref.	Ref.	Ref.	Ref.	Ref.
Any Medicaid	0.85 (0.63, 1.16)	0.58 (0.39, 0.86)	0.64 (0.44, 0.94)	0.65 (0.44, 0.94)	0.74 (0.51, 1.08)	0.59 (0.41, 0.84)	0.56 (0.38, 0.82)
Other insurance	0.61 (0.45, 0.83)	0.51 (0.32, 0.82)	0.59 (0.38, 0.92)	0.51 (0.34, 0.78)	0.63 (0.41, 0.96)	0.46 (0.31, 0.68)	0.50 (0.32, 0.80)
Uninsured	0.44 (0.26, 0.74)	0.25 (0.13, 0.48)	0.60 (0.33, 1.07)	0.29 (0.16, 0.52)	0.36 (0.19, 0.68)	0.35 (0.21, 0.59)	0.24 (0.14, 0.42)
Education of mother							
<12 years	0.57 (0.40, 0.81)	0.93 (0.57, 1.54)	0.83 (0.51, 1.35)	0.77 (0.49, 1.22)	1.09 (0.69, 1.73)	0.72 (0.46, 1.11)	0.64 (0.40, 1.04)
12 years	0.63 (0.48, 0.83)	0.81 (0.55, 1.18)	0.86 (0.60, 1.24)	0.88 (0.61, 1.27)	0.74 (0.51, 1.07)	0.92 (0.66, 1.28)	0.71 (0.49, 1.02)
>12 years, non-college grad	0.65 (0.50, 0.83)	0.97 (0.66, 1.42)	0.86 (0.61, 1.20)	0.88 (0.59, 1.32)	1.06 (0.75, 1.50)	0.87 (0.64, 1.18)	0.67 (0.47, 0.97)
College grad	Ref.	Ref.	Ref.	Ref.	Ref.	Ref.	Ref.
Provider facility type							
All private	Ref.	Ref.	Ref.	Ref.	Ref.	Ref.	Ref.
Mixed	0.91 (0.69, 1.20)	1.08 (0.65, 1.80)	1.02 (0.64, 1.62)	1.24 (0.85, 1.80)	0.95 (0.60, 1.52)	1.11 (0.72, 1.71)	1.00 (0.66, 1.49)
All military/other	0.85 (0.55, 1.32)	0.89 (0.48, 1.64)	0.60 (0.33. 1.09)	0.79 (0.47, 1.32)	0.93 (0.52, 1.67)	0.57 (0.33, 0.98)	0.82 (0.46, 1.49)
All public	0.68 (0.53, 0.88)	0.53 (0.37, 0.77)	0.76 (0.52, 1.10)	0.54 (0.39, 0.74)	0.53 (0.38, 0.75)	0.78 (0.56, 1.09)	0.46 (0.33, 0.64)
All hospital	0.56 (0.44, 0.72)	0.38 (0.27, 0.53)	0.39 (0.28, 0.54)	0.41(0.30, 0.56)	0.45 (0.33, 0.61)	0.43 (0.32, 0.60)	0.38 (0.27, 0.54)
Census region							
Northeast	Ref.	Ref.	Ref.	Ref.	Ref.	Ref.	Ref.
Midwest	0.83 (0.66, 1.05)	0.81 (0.57, 1.17)	1.07 (0.75, 1.54)	0.72 (0.52, 0.99)	0.96 (0.69, 1.33)	1.00 (0.72, 1.38)	0.81 (0.58, 1.14)
South	0.87 (0.69, 1.10)	0.73 (0.51, 1.05)	0.86 (0.60, 1.22)	0.68 (0.49, 0.95)	0.78 (0.56, 1.09)	0.88 (0.63, 1.21)	0.72 (0.51, 1.02)
West	0.57 (0.43, 0.75)	0.42 (0.28, 0.63)	0.52 (0.35, 0.78)	0.46 (0.32, 0.66)	0.48 (0.33, 0.69)	0.48 (0.34, 0.70)	0.50 (0.34, 0.74)
Race/ethnicity							
Non-hispanic white only	Ref.	Ref.	Ref.	Ref.	Ref.	Ref.	Ref.
Non-hispanic black only	0.87 (0.67, 1.14)	0.81 (0.56, 1.18)	1.02 (0.71, 1.48)	0.97 (0.68, 1.37)	0.82 (0.57, 1.18)	1.07 (0.75, 1.52)	0.93 (0.63, 1.38)
Hispanic	1.15 (0.91, 1.44)	1.15 (0.84, 1.58)	1.34 (0.96, 1.86)	1.40 (1.05, 1.87)	1.15 (0.84, 1.57)	1.46 (1.07, 2.00)	1.32 (0.96, 1.81)
Non-hispanic other + multiple race	0.99 (0.79, 1.24)	0.90 (0.63, 1.27)	1.03 (0.74, 1.44)	1.00 (0.73, 1.36)	1.06 (0.77, 1.46)	0.86 (0.63, 1.18)	0.85 (0.61, 1.18)

Note: (1) Odds ratios (ORs) and 95% confidence intervals from survey-weighted logistic regression accounting for stratification and clustering. (2) Table 3 presents odds ratios for six explanatory variables (income-to-poverty ratio, health insurance, maternal education, provider type, census region, and race/ethnicity). Models additionally adjusted for WIC participation, child age, sex, maternal age, and marital status.

**Table 4 vaccines-13-01256-t004:** Concentration Indices for Socioeconomic Disparities in Receipt of the 7-Vaccine Series Before and After Adjustment for Covariates.

Variable	Vaccination: 7-Vaccine Series (4:3:1:3:3:1:3)			
	CI, Unadjusted	*p* Value	CI, Adjusted	*p* Value
Income to poverty (FPL) category	0.04	<0.01	0.02	<0.01
Health insurance type	0.03	<0.01	0.01	<0.01
Education of mother	0.04	<0.01	0.02	<0.01
Provider facility type	0.03	<0.01	0.02	<0.01
Census region	<0.01	0.57	<0.01	0.78
Race/ethnicity	0.01	0.04	<0.01	0.72

**Table 5 vaccines-13-01256-t005:** Unadjusted Concentration Indices for Key Explanatory Variables Across Seven Childhood Vaccines.

Variable	CI, Adjusted													
	DTP4		Pol3		MCV1		Hib3		HepB3		VCV1		PCV3	
Income to poverty (FPL) category	0.02	<0.01	0.01	<0.01	<0.01	0.36	0.01	<0.01	0.01	<0.01	<0.01	0.46	0.01	<0.01
Health insurance type	0.01	<0.01	0.01	<0.01	0.01	0.02	0.01	<0.01	<0.01	0.20	0.01	0.01	0.01	<0.01
Education of mother	0.02	<0.01	0.01	0.02	<0.01	0.07	0.01	<0.01	<0.01	0.08	0.01	0.01	0.01	<0.01
Provider facility type	0.02	<0.01	0.01	<0.01	0.01	<0.01	0.02	<0.01	0.01	<0.01	0.01	<0.01	0.01	<0.01
Census region	<0.01	0.22	<0.01	0.35	<0.01	0.28	<0.01	0.41	<0.01	0.27	<0.01	0.55	<0.01	0.23
Race/ethnicity	<0.01	0.42	<0.01	0.52	<0.01	0.34	<0.01	0.74	<0.01	0.71	<0.01	0.99	<0.01	0.92

## Data Availability

The original data presented in the study are openly available in the 2023 National Immunization Survey–Child (NIS-Child) at https://www.cdc.gov/nis/php/datasets-child/index.html (accessed on 1 September 2025).

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
