# Peer review of "Socioeconomic Disparities in Childhood Vaccination Coverage in the United States: Evidence from a Post-COVID-19 Birth Cohort"

_vaccines, 2025, doi:10.3390/vaccines13121256_

Round 1

Reviewer 1 Report

Comments and Suggestions for Authors

This manuscript addresses the persistence of socioeconomic disparities in routine childhood vaccination in a post–COVID-19 birth cohort. Given the pandemic’s documented disruptions to preventive services and the rise in vaccine hesitancy, updated evidence on equity gaps is highly relevant for policy, program planning, and targeted interventions.

Major comments

  • Page 1, lines 14–20: The objective is stated but could be more explicitly framed as a gap in post-pandemic evidence. Clarifying novelty would strengthen the introduction. The authors would benefit from the findings of the following tw high-level papers on the global burden of childhood vaccination.
  • Haeuser, E., Byrne, S., Nguyen, J., Raggi, C., McLaughlin, S.A., Bisignano, C., Harris, A.A., Smith, A.E., Lindstedt, P.A., Smith, G. and Herold, S.J., 2025. Global, regional, and national trends in routine childhood vaccination coverage from 1980 to 2023 with forecasts to 2030: a systematic analysis for the Global Burden of Disease Study 2023. The Lancet.
  • Brown A, Ahmed MB, Gebremeskel TG, Naik GR. The unfinished agenda of communicable diseases among children and adolescents before the COVID-19 pandemic, 1990–2019: a systematic analysis of the Global Burden of Disease Study 2019. The Lancet. 2023 Jul 22;402(10398):313-35.
  • GBD 2021 US Burden of Disease and Forecasting Collaborators. Burden of disease scenarios by state in the USA, 2022-50: a forecasting analysis for the Global Burden of Disease Study 2021. Lancet. 2024 Dec 7;404(10469):2341-2370. doi: 10.1016/S0140-6736(24)02246-3. Erratum in: Lancet. 2025 May 3;405(10489):1579. doi: 10.1016/S0140-6736(25)00828-1. PMID: 39645377; PMCID: PMC11715278.

- Page 2, lines 87–96: The sampling strategy and exclusion criteria are briefly described. Please specify how missing data were handled and whether sensitivity analyses were performed.

- Page 3, lines 127–158: The construction of ranking variables for categorical CI analyses requires clearer justification. The current explanation may be difficult for non-technical readers.

- Page 7–10: While effect sizes are accurately presented, the discussion should better reconcile the presence of statistically significant disparities with their very small absolute magnitude—particularly for CIs near zero.

- Page 11, lines 268–282: Consider elaborating on policy implications, e.g., which socioeconomic groups would most benefit from targeted efforts. The earlier suggested readings are very beneficial in addition to Nashwan AJ, Abuhammad S. Zero-Dose Children, Misinformation, and Vaccine Hesitancy. Cureus. 2025 Apr 10;17(4):e82028. doi: 10.7759/cureus.82028. PMID: 40351967; PMCID: PMC12065619. 

Minor comments

- Tables 1–5: Some percentages appear misaligned (e.g., Table 1 FPL categories). Ensure consistent decimal formatting.

- Page 1, line 41: Replace “modest in scale” with a quantified interpretation aligned with CI values for precision.

- Page 2, lines 74–76: A brief definition or citation regarding spillover hesitancy would aid clarity.

- Please proofread for small typographical inconsistencies (e.g., missing parentheses in Table 3: “0.6(0.33. 1.09)”).

Comments on the Quality of English Language

The manuscript is generally well written; however, several sections could benefit from minor polishing of the English language to improve clarity and readability. In particular, some sentences are lengthy and could be streamlined for better flow (e.g., Pages 2–3 when describing disparities and ranking procedures). Additionally, a few typographical inconsistencies and formatting issues appear in the tables (such as misplaced parentheses and inconsistent decimal formatting). Refining these elements will enhance the overall quality of the presentation without altering the scientific content. If available, a light professional language edit is recommended.

Author Response

This manuscript addresses the persistence of socioeconomic disparities in routine childhood vaccination in a post–COVID-19 birth cohort. Given the pandemic’s documented disruptions to preventive services and the rise in vaccine hesitancy, updated evidence on equity gaps is highly relevant for policy, program planning, and targeted interventions.

Major comments

Comments 1:

  • Page 1, lines 14–20: The objective is stated but could be more explicitly framed as a gap in post-pandemic evidence. Clarifying novelty would strengthen the introduction. The authors would benefit from the findings of the following tw high-level papers on the global burden of childhood vaccination.
  • Haeuser, E., Byrne, S., Nguyen, J., Raggi, C., McLaughlin, S.A., Bisignano, C., Harris, A.A., Smith, A.E., Lindstedt, P.A., Smith, G. and Herold, S.J., 2025. Global, regional, and national trends in routine childhood vaccination coverage from 1980 to 2023 with forecasts to 2030: a systematic analysis for the Global Burden of Disease Study 2023. The Lancet.
  • Brown A, Ahmed MB, Gebremeskel TG, Naik GR. The unfinished agenda of communicable diseases among children and adolescents before the COVID-19 pandemic, 1990–2019: a systematic analysis of the Global Burden of Disease Study 2019. The Lancet. 2023 Jul 22;402(10398):313-35.
  • GBD 2021 US Burden of Disease and Forecasting Collaborators. Burden of disease scenarios by state in the USA, 2022-50: a forecasting analysis for the Global Burden of Disease Study 2021. Lancet. 2024 Dec 7;404(10469):2341-2370. doi: 10.1016/S0140-6736(24)02246-3. Erratum in: Lancet. 2025 May 3;405(10489):1579. doi: 10.1016/S0140-6736(25)00828-1. PMID: 39645377; PMCID: PMC11715278.

Response 1:

Thank you for this insightful comment. We have revised the Introduction to more explicitly frame the study objective as addressing a gap in post–COVID-19 evidence on socioeconomic disparities in childhood vaccination in the United States. We have incorporated key insights from the suggested high-level papers to contextualize our study within broader global and national trends. Specifically:

We cite Haeuser et al. (2025) to highlight that while global routine childhood vaccination coverage nearly doubled from 1980 to 2023, progress slowed in the decade before the pandemic, and the COVID-19 pandemic led to sharp declines in coverage for many vaccines with levels not yet returned to pre-pandemic baselines as of 2023.

We cite Brown et al. (2023) to acknowledge that before the pandemic, communicable diseases remained a substantial burden among children and adolescents globally, with persistent inequities across socioeconomic strata. Their analysis underscores the need for continued policy focus on vaccine-preventable diseases, especially among younger children in low-resource settings.

We cite the GBD 2021 US Burden of Disease and Forecasting Collaborators (2024) to situate our study within the US-specific burden landscape. Their forecasts indicate that improving childhood vaccination coverage is one of the key interventions that could substantially improve US life expectancy and healthy life expectancy by 2050.

Our study builds on these global and national assessments by providing timely, US-specific evidence on socioeconomic disparities in routine childhood vaccination in the post-pandemic period, a gap not yet addressed in the literature. The revised text appears in the Introduction section (Page 2, lines 94–101).

Comments 2:- Page 2, lines 87–96: The sampling strategy and exclusion criteria are briefly described. Please specify how missing data were handled and whether sensitivity analyses were performed.

Response 2:Thank you for this important clarification. As mentioned in method, we restricted the analytic sample to children with non-missing vaccination outcomes and positive provider-verified sampling weights (PROVWT_C). Children with missing data on key explanatory variables or covariates were excluded from the analysis. This approach is consistent with NIS-Child analytical guidelines. Household income was measured using the income-to-poverty ratio, for which we used the NIS-Child–provided imputed variable generated by CDC using standardized multiple imputation procedures, thereby reducing bias due to missing income information. We acknowledge this as a limitation in the Discussion and note that the exclusion of participate with missing data may introduce selection bias. We have added a sentence in the Discussion section (Page 13, lines 403–408) to explicitly state this.

Sensitivity analyses were performed by comparing unadjusted and adjusted (standardized) concentration indices across all disparity dimensions examined, including poverty status, insurance type, maternal education, provider facility type, census region, and race/ethnicity. The consistency in the direction and relative magnitude of estimates across specifications indicated that the findings were not sensitive to model assumptions. These clarifications have been added to the Methods (Page 4, lines 185-187) and Results (Page 11, lines 325-327) sections.

Comments 3:- Page 3, lines 127–158: The construction of ranking variables for categorical CI analyses requires clearer justification. The current explanation may be difficult for non-technical readers.

Response 3:We thank the reviewer for this helpful suggestion. We agree that the original description was overly technical and may not have been sufficiently accessible to non-technical readers. In the revised text, we explicitly explain that concentration indices require an ordered socioeconomic ranking, that several variables (e.g., insurance type, provider type, region, and race/ethnicity) lack an inherent order, and that the income-to-poverty ratio was therefore used as an external reference to establish a meaningful socioeconomic ordering. The income-to-poverty ratio provides a direct, widely used, and interpretable measure of household material resources and so-cioeconomic advantage, and is therefore well suited as a unifying ordering metric. (Page 4, lines 169–178)

Comments 4:- Page 7–10: While effect sizes are accurately presented, the discussion should better reconcile the presence of statistically significant disparities with their very small absolute magnitude—particularly for CIs near zero.

Response 4:Thank you for this important observation. We acknowledge that the bounded nature of the concentration index for binary outcomes with high means—such as vaccination coverage exceeding 90%—mathematically constrains the possible range of CI values, leading to numerically small indices even when relative inequalities are present and statistically significant. We reference methodological literature on the interpretation of CIs for binary outcomes with high means. As follows:

Wagstaff, A. The bounds of the concentration index when the variable of interest is binary, with an application to immunization inequality. Health Econ 2005, 14, 429-432, doi:10.1002/hec.953.

Nevertheless, even small absolute disparities can have meaningful public health implications if they persist across large populations or over time, potentially leading to clusters of under-immunization and increased outbreak risk in disadvantaged communities. In the Discussion (Page 12, lines 351–354), we have added a paragraph explicitly addressing this issue.  

Comments 5:- Page 11, lines 268–282: Consider elaborating on policy implications, e.g., which socioeconomic groups would most benefit from targeted efforts. The earlier suggested readings are very beneficial in addition to Nashwan AJ, Abuhammad S. Zero-Dose Children, Misinformation, and Vaccine Hesitancy. Cureus. 2025 Apr 10;17(4):e82028. doi: 10.7759/cureus.82028. PMID: 40351967; PMCID: PMC12065619. 

Response 5:We thank the reviewer for this constructive suggestion. We have expanded the policy implications section to more clearly identify priority groups that may benefit most from targeted interventions, including children from low-income households, those insured by Medicaid or uninsured, children of mothers with lower educational attainment, and those receiving care primarily from public or hospital-based providers (Page 12, Lines 383-386).

Minor comments

Comments 6:- Tables 1–5: Some percentages appear misaligned (e.g., Table 1 FPL categories). Ensure consistent decimal formatting.

Response 6:We thank the reviewer for the careful attention to detail. We have corrected formatting inconsistencies, including alignment of percentages and decimal places. Household income relative to the federal poverty level (FPL), categorized as <1×FPL, 1–<2×FPL, and 2–≤3×FPL.

Comments 7:- Page 1, line 41: Replace “modest in scale” with a quantified interpretation aligned with CI values for precision.

Response 7:Thank you for your suggestion. We have revised the sentence in the Abstract as follows: These findings suggest that while inequities remain statistically measurable, their scale is limited in absolute terms.

Comments 8:- Page 2, lines 74–76: A brief definition or citation regarding spillover hesitancy would aid clarity.

Response 8:Thank you for your suggestion to clarify the term “spillover hesitancy.” We agree that the phrasing could be ambiguous, as “spillover hesitancy” is not a formally defined term in the literature. To avoid potential misinterpretation and to enhance clarity, we have revised the sentence to more directly describe the phenomenon evidenced in the cited study. (Page 2, Lines 88)

Comments 9:- Please proofread for small typographical inconsistencies (e.g., missing parentheses in Table 3: “0.6(0.33. 1.09)”).

Response 9:Thank you. We have thoroughly proofread the manuscript and corrected the errors in Table 3.

Comments on the Quality of English Language

Comments 10:The manuscript is generally well written; however, several sections could benefit from minor polishing of the English language to improve clarity and readability. In particular, some sentences are lengthy and could be streamlined for better flow (e.g., Pages 2–3 when describing disparities and ranking procedures). Additionally, a few typographical inconsistencies and formatting issues appear in the tables (such as misplaced parentheses and inconsistent decimal formatting). Refining these elements will enhance the overall quality of the presentation without altering the scientific content. If available, a light professional language edit is recommended.

Response 10:We thank the reviewer for their positive feedback on the manuscript and for their helpful suggestions regarding language clarity and presentation.

In response to the comments, we have carefully reviewed the entire manuscript to improve readability. We have streamlined several lengthy sentences, particularly in the Methods section (Pages 2–3) where disparities and ranking procedures were described, to enhance clarity and conciseness. All tables have been thoroughly checked and corrected for typographical inconsistencies, including misplaced parentheses and decimal formatting. Numerical values are now uniformly presented with two decimal places, and confidence intervals are consistently formatted.

Reviewer 2 Report

Comments and Suggestions for Authors

This manuscript presents a timely and important analysis of socioeconomic disparities in childhood vaccination coverage in the United States, focusing on a birth cohort of children born during or after the COVID-19 pandemic. Overall, the manuscript is well-structured, the methods are robust, and the results are clearly presented. However, I have a few observations to further clarify the findings and strengthen the discussion:

  1. The Methods section specifies that the analysis focuses on children born in 2021-2022, representing those reaching age 19–35 months during or after the COVID-19 pandemic. Please confirm if the stated birth cohort of 2021-2022 in the Abstract is precisely the group selected, or if the standard NIS-Child age range (19–35 months at the time of the survey) was used, which would capture some children born in 2020 as well.
  2. The approach to deriving a rank variable for categorical/nominal disparity dimensions (e.g., insurance type, provider type, region, race/ethnicity) by sorting groups based on their weighted mean income-to-poverty ratio is novel. Please add a brief justification in Section 2.6.2. Equity Analysis: Adjusted Concentration Indices for why the income-to-poverty ratio was chosen as the basis for ordering these groups, as opposed to, for example, weighted mean maternal education.
  3. The Discussion should be slightly expanded to specifically frame how these results compare to pre-pandemic data. The paper notes that the patterns are consistent with prior research , but an explicit statement on whether the disparities appear to have been exacerbated, reduced, or remained stable (in terms of OR/CI magnitude) compared to historical NIS-Child cohorts would significantly enhance the value of assessing this post-COVID-19 cohort.

Author Response

Response Letter to Editor and Reviewers

Reviewer 2

This manuscript presents a timely and important analysis of socioeconomic disparities in childhood vaccination coverage in the United States, focusing on a birth cohort of children born during or after the COVID-19 pandemic. Overall, the manuscript is well-structured, the methods are robust, and the results are clearly presented. However, I have a few observations to further clarify the findings and strengthen the discussion:

Comments 1:The Methods section specifies that the analysis focuses on children born in 2021-2022, representing those reaching age 19–35 months during or after the COVID-19 pandemic. Please confirm if the stated birth cohort of 2021-2022 in the Abstract is precisely the group selected, or if the standard NIS-Child age range (19–35 months at the time of the survey) was used, which would capture some children born in 2020 as well.

Response 1:Thank you for this important clarification. You are correct that the standard NIS-Child sampling frame includes children aged 19–35 months at the time of interview. In our current analysis, we used the 2023 NIS-Child provider-verified sample within the standard 19–35 month age range, rather than restricting the sample to a birth-year cohort of 2021–2022. Therefore, the analytic sample can include some children born in 2020 as well.

We agree that the wording in the Abstract could be interpreted as a strict birth-cohort restriction. We have revised the Abstract to accurately describe the sample as children aged 19–35 months in the 2023 NIS-Child survey, thereby avoiding any ambiguity about inclusion of 2020 births. (Page 1, lines 18–20)

Comments 2:The approach to deriving a rank variable for categorical/nominal disparity dimensions (e.g., insurance type, provider type, region, race/ethnicity) by sorting groups based on their weighted mean income-to-poverty ratio is novel. Please add a brief justification in Section 2.6.2. Equity Analysis: Adjusted Concentration Indices for why the income-to-poverty ratio was chosen as the basis for ordering these groups, as opposed to, for example, weighted mean maternal education.

Response 2:We have added a justification in Section 2.6.2 explaining why the income-to-poverty ratio was used to order categorical or nominal disparity dimensions.

Specifically, concentration indices are designed to quantify inequality along a socioeconomic gradient, which requires an ordered rank that reflects relative socioeconomic position. The income-to-poverty ratio provides a direct, widely used, and interpretable measure of household material resources and socioeconomic advantage, and is therefore well suited as a unifying ordering metric.

We did not use maternal education for this purpose because maternal education is itself a key disparity dimension examined in our analyses. Using it as the basis for ordering other dimensions would introduce conceptual overlap and potential endogeneity.

These clarifications have been added to Section 2.6.2 of the revised manuscript.(Page 4, lines 169–178)

Comments 3:The Discussion should be slightly expanded to specifically frame how these results compare to pre-pandemic data. The paper notes that the patterns are consistent with prior research , but an explicit statement on whether the disparities appear to have been exacerbated, reduced, or remained stable (in terms of OR/CI magnitude) compared to historical NIS-Child cohorts would significantly enhance the value of assessing this post-COVID-19 cohort.

Response 3:We thank the reviewer for this constructive suggestion. We agree that more explicitly situating our findings relative to pre-pandemic NIS-Child evidence strengthens the interpretation of this post–COVID-19 cohort. Specifically, we added text noting that our findings are consistent with established pre-pandemic NIS-Child evidence documenting lower vaccination coverage among children living below the poverty level, with reported gaps ranging from 2.7 percentage points for ≥1 dose of MMR to as high as 19.9 percentage points for ≥2 doses of influenza vaccine, depending on the vaccine examined. We state that the socioeconomic disparities observed in our analysis appear to have persisted into the post–COVID-19 period rather than showing clear evidence of exacerbation or reduction in magnitude. These additions appear on Page 13, Lines 375–381.